# Effect of INTELLiVENT-ASV versus Conventional Ventilation on Ventilation Intensity in Patients with COVID-19 ARDS—An Observational Study

**DOI:** 10.3390/jcm10225409

**Published:** 2021-11-19

**Authors:** Laura A. Buiteman-Kruizinga, Hassan E. Mkadmi, Ary Serpa Neto, Matthijs D. Kruizinga, Michela Botta, Marcus J. Schultz, Frederique Paulus, Pim L.J. van der Heiden

**Affiliations:** 1Department of Intensive Care, Reinier de Graaf Hospital, 2625 AD Delft, The Netherlands; pim.vanderheiden@rdgg.nl; 2Department of Intensive Care, Amsterdam University Medical Centers, Location ‘AMC’, 1105 AZ Amsterdam, The Netherlands; ary.serpaneto@monash.edu (A.S.N.); m.botta@amsterdamumc.nl (M.B.); marcus.j.schultz@gmail.com (M.J.S.); f.paulus@amsterdamumc.nl (F.P.); 3Department of Research, Reinier de Graaf Hospital, 2625 AD Delft, The Netherlands; Hassan.mkadmi@gmail.com; 4Australian and New Zealand Intensive Care–Research Centre (ANZIC–RC), Monash University, Melbourne, VIC 3004, Australia; 5Department of Critical Care Medicine, Hospital Israelita Albert Einstein, São Paulo 05652-900, Brazil; 6Department of Pediatrics, Juliana Children’s Hospital, 2545 AA The Hague, The Netherlands; M.Kruizinga@hagaziekenhuis.nl; 7Laboratory of Experimental Intensive Care and Anesthesiology (LEICA), Amsterdam University Medical Centers, Location ‘AMC’, 1105 AZ Amsterdam, The Netherlands; 8Mahidol–Oxford Tropical Medicine Research Unit (MORU), Mahidol University, Bangkok 10400, Thailand; 9Nuffield Department of Medicine, University of Oxford, Oxford OX3 7FZ, UK; 10ACHIEVE, Centre of Applied Research, Faculty of Health, Amsterdam University of Applied Sciences, 1105 AZ Amsterdam, The Netherlands

**Keywords:** COVID-19, ARDS, automated ventilation, closed-loop ventilation, INTELLiVENT-ASV, intensity of ventilation, mechanical power, driving pressure

## Abstract

Driving pressure (ΔP) and mechanical power (MP) are associated with outcomes in critically ill patients, irrespective of the presence of Acute Respiratory Distress Syndrome (ARDS). INTELLiVENT-ASV, a fully automated ventilatory mode, controls the settings that affect ΔP and MP. This study compared the intensity of ventilation (ΔP and MP) with INTELLiVENT-ASV versus conventional ventilation in a cohort of COVID-19 ARDS patients in two intensive care units in the Netherlands. The coprimary endpoints were ΔP and MP before and after converting from conventional ventilation to INTELLiVENT-ASV. Compared to conventional ventilation, INTELLiVENT-ASV delivered ventilation with a lower ΔP and less MP. With conventional ventilation, ΔP was 13 cmH_2_O, and MP was 21.5 and 24.8 J/min, whereas with INTELLiVENT-ASV, ΔP was 11 and 10 cmH_2_O (mean difference –2 cm H2O (95 %CI –2.5 to –1.2 cm H_2_O), *p* < 0.001) and MP was 18.8 and 17.5 J/min (mean difference –7.3 J/Min (95% CI –8.8 to –5.8 J/min), *p* < 0.001). Conversion from conventional ventilation to INTELLiVENT-ASV resulted in a lower intensity of ventilation. These findings may favor the use of INTELLiVENT-ASV in COVID-19 ARDS patients, but future studies remain needed to see if the reduction in the intensity of ventilation translates into clinical benefits.

## 1. Introduction

Limiting the intensity of ventilation could improve outcomes in patients with acute respiratory distress syndrome (ARDS) [1,2,3]. This approach may also benefit patients with coronavirus disease 2019 (COVID-19) ARDS [4]. The intensity of ventilation is reflected by multiple parameters. The first is the driving pressure (ΔP), i.e., the pressure applied by the ventilator to support the delivery of a tidal volume (V_T_) and, as such, represents the strain applied to the lung with each breath during mechanical ventilation [5]. The second is the mechanical power of ventilation (MP)—the energy used to overcome airway resistance and respiratory system compliance, part of which acts directly on lung tissue [6,7]. The latter measure combines multiple ventilatory parameters, including V_T_ and ΔP, but also respiratory rate (RR) [8,9].

It can be very challenging, if not practically impossible, to keep the intensity of ventilation low at all times. INTELLiVENT-Adaptive Support Ventilation (ASV) is a fully automated, closed-loop ventilatory mode that automatically controls gas exchange at the lowest work of breathing [10] and at the lowest force of breathing [11]. This means that both ΔP and MP are, at least in part, under the control of INTELLiVENT-ASV.

It is uncertain whether INTELLiVENT-ASV affects the intensity of ventilation in COVID-19 patients with ARDS. The aim of this substudy of the ‘PRactice of VENTilation in COVID-19’ (PRoVENT–COVID) study [12] was to compare ΔP and MP with INTELLiVENT-ASV versus conventional ventilation. We hypothesized the intensity of ventilation to decrease after conversion from conventional non-automated ventilation to INTELLiVENT-ASV.

## 2. Materials and Methods

### 2.1. Study Design

This was a retrospective 2-center substudy within a large national observational study undertaken during the first wave of the COVID-19 pandemic in the Netherlands. The substudy was conducted at the Intensive Care Unit (ICU) of the Reinier de Graaf Hoss alignmentpital in Delft and the ICU of the Amsterdam University Medical Centers, ‘location AMC’ in Amsterdam, the Netherlands. The study protocol (number W20_157#20.171) was approved on 7 April 2020 by the Institutional Review Board of the AMC (chairperson Prof. Dr. J.A. Swinkels), Amsterdam, and was prepublished [13], and the study was registered at www.clinicaltrials.gov (accessed on 15 April 2020); trial identification number NCT04346342). The need for informed consent was waived because of the observational nature of this study and because the decision to use conventional ventilation, or INTELLiVENT-ASV, was left to the discretion of attending physicians and nurses and in accordance with the local guideline for ventilation in the 2 ICUs. Before and after the conversion to INTELLiVENT-ASV, fairly identical target ranges for end-tidal carbon dioxide (etCO_2_) and peripheral oxygen saturation (SpO_2_) were used. We targeted normocapnia and normoxemia with both ventilation modes in all patients. Pressure limits were left unchanged. Additional details on ventilator settings are depicted in Table 1. A statistical analysis plan for this substudy, written and finalized before closing the database, was reported at the study website [14] and is available in the online supplement.

### 2.2. Study Population

Patients aged 18 years or older with COVID-19 confirmed with RT-PCR for SARS-CoV-2 and ARDS according to the Berlin definition [15] were eligible if they received invasive pressure-controlled ventilation in one of the 2 participating ICUs, had received at least 3 h of conventional ventilation before converting to INTELLiVENT-ASV and at least 3 subsequent hours of INTELLiVENT-ASV. Patients were excluded if the conversion from conventional ventilation to INTELLiVENT-ASV was not initiated within the first 3 days of invasive ventilation, when there was a change in body position, e.g., from prone or supine or vice versa, or when there was spontaneous breathing activity at any timepoint during the timeframe of data collection. Spontaneous breathing was determined when comparing set respiratory rate with observed respiratory rate, and if the latter was >2 higher, it was seen as evidence for the presence of spontaneous breathing activity.

### 2.3. Collected Data

The severity of illness, medication and vital signs were obtained at baseline. Ventilation variables and parameters were collected at 4 consecutive timepoints: at 2 and 1 h before and at 1 and 2 h after conversion from conventional ventilation to INTELLiVENT-ASV. Thus, we had a maximum of 4 timepoints at which ΔP and MP could be calculated. ΔP was calculated as plateau pressure (Pplat) minus positive end-expiratory pressure (PEEP). MP was calculated as [6]:MP (J/min): 0·098 ∗ RR ∗ V_T_ ∗ (Peak pressure (Ppeak) − (0·5 ∗ ΔP) (1)

### 2.4. Outcomes

The coprimary outcomes were ΔP and MP before and after conversion from conventional ventilation to INTELLiVENT-ASV. Secondary outcomes were other key ventilation variables and parameters, including V_T_, PEEP, Pmax and RR, at the same timepoints before and after conversion.

### 2.5. Statistical Analysis

Descriptive statistics were used to describe the study population, and data were expressed in number and relative proportions for categorical variables and median (quartile 25%–quartile 75%) or mean (±SD) for continuous variables. Proportions were compared using the chi-squared test or Fisher exact test as required by variable distribution, and continuous variables were compared using the Wilcoxon Rank Sum Test or the Wilcoxon signed-rank test as appropriate. Effects are presented with a 95% confidence interval (95% CI).

A mixed-effects generalized linear model with a Gaussian distribution was used, wherein ventilation mode was used as a fixed effect and patients as a random effect, to account for repeated measurements.

To compare ΔP, MP, V_T_, PEEP, Pmax, RR and other ventilator parameters with INTELLiVENT-ASV versus conventional ventilation, cumulative distribution plots were constructed. Medians were compared using the Wilcoxon signed-rank test. In addition, the relation between V_T_ and ΔP at the 4 timepoints was visualized in plots using least square method regression analysis. Scatterplots and line graphs were also used to show how individual changes in V_T_ related to changes in ΔP.

We performed 2 post hoc analyses. In the first post hoc analysis, MP was calculated using another equation than the one proposed above as [16]:MP (J/min) = 0·098 ∗ RR ∗ V_T_ ∗ (Pinsp + PEEP) (2)

In the second post hoc analysis, MP was normalized for respiratory system compliance (C_RS_) and was calculated as:C_RS_ (mL/cmH_2_O) = V_T_/(Pplat − PEEP), andMP_NORM_ (J/min per mL/cmH_2_O) = MP/C_RS_(3)

There were no missing data. A *p* value < 0.05 was considered significant. Analyses were performed with SPSS version 25 (descriptive statistics, comparison of ventilation parameters) and R version 3.6.3 (generalized linear mixed-effects model).

## 3. Results

### 3.1. Patients

Between 1 March and 1 June 2020, 144 patients were screened for eligibility (Figure 1). A total of 94 were not enrolled: 8 patients that did not receive invasive ventilation, 32 that were never connected to a ventilator that can provide INTELLiVENT-ASV and 43 that did not receive INTELLiVENT-ASV within the timeframe of interest. Two additional patients were excluded because they received INTELLiVENT-ASV for less than 3 h after the conversion to INTELLIVENT-ASV, two because of a change in body position and seven because of spontaneous breathing activity within the timeframe of data collection.

Demographic data, including preceding medication, the severity of disease and comorbidities, are presented in Table 2. The majority of patients were male, and the median age was 63 (IQR 51 to 69) years. Of all included patients, 14% met the current definition for mild ARDS, and 55% and 31% met the definition for moderate or severe ARDS, respectively. Patients were under invasive ventilation for a median of 10 h (IQR 3 to 48) before ventilation was converted from conventional ventilation to INTELLiVENT-ASV.

### 3.2. Intensity of Ventilation

At 2 and 1 h before the conversion from conventional ventilation to INTELLiVENT-ASV, the median ΔP was 13 (IQR 10 to 17) and 13 (IQR 10 to 17) cmH_2_O, the median MP was 21.5 (IQR 14.6 to 32.1) and 24.8 (IQR 19.4 to 31) J/min. 1 and 2 h after the conversion, the median ΔP was 11 (IQR 9 to 14) and 10 (IQR 8 to 14) cmH_2_O (mean difference –2 cmH_2_O (95% CI –2.5 to –1.2 cm H_2_O); *p* < 0.001) and the median MP was 18.8 (IQR 12.2 to 22) and 17.5 (IQR 12.2 to 21.1) J/min (mean difference of –7.7 J/min (95% CI –8.8 to –5.8 J/min); *p* < 0.001) (Figure 2).

### 3.3. Other Ventilation Variables and Parameters

Conversion from conventional ventilation to INTELLiVENT-ASV did not result in a change in median PEEP. The conversion was associated with a small increase in median V_T_, but most patients maintained a V_T_ of < 8 mL/kg PBW. Before conversion to INTELLiVENT-ASV, 7 of 51 patients (14%) received ventilation with a VT > 8 mL/kg PBW. At 1 and 2 h after conversion, 10 (19%) and 8 patients (16%) received ventilation with a VT of > 8 mL/kg PBW. With INTELLiVENT-ASV, when V_T_ increased, ΔP decreased. In addition, when ΔP was high with conventional ventilation, V_T_ decreased with INTELLiVENT-ASV (Figure 3Figure 4 and Figure 5 and Appendix A). Median RR, Pmax, minute volume and FiO_2_ decreased with the conversion from conventional ventilation to INTELLiVENT-ASV (Table 2 and Appendix A). Compliance of the respiratory system improved while patients were ventilated with INTELLiVENT-ASV (Table 3 and Appendix A). Conversion to INTELLiVENT-ASV did not affect the etCO_2_ and SpO_2_ values (Figure 5 and Table 3).

### 3.4. Post hoc Analyses

Neither using an alternate equation (Appendix A) nor normalizing MP (Appendix A) changed the findings that converting to INTELLiVENT-ASV reduces the intensity of ventilation.

## 4. Discussion

The findings of this study in COVID-19 patients with ARDS show that converting from non-automated ventilation to the automated ventilatory mode INTELLiVENT-ASV reduces the intensity of ventilation, as reflected by (1) a reduction in ΔP and (2) a reduction in MP. Limiting ΔP and MP have been proposed as targets that may result in better outcomes in patients with ARDS [1,2,3], and automated ventilation could be one practical way to achieve these goals.

This study has strengths and limitations. First, this study was performed in ICUs with physicians and nurses with extensive experience in the use of lung-protective ventilation and also the use of INTELLiVENT-ASV. The first can be seen as a strength, as this means that we compared ‘best practice’ in lung-protective ventilation during conventional ventilation with fully automated ventilation. However, the second could be seen as a limitation, as this may reduce the generalizability of the findings. Of note, adequate input into the ventilator by the caregivers remains necessary for the optimal use of INTELLiVENT-ASV, and the quality of input increases with experience. Other strengths are that we strictly followed a predefined analysis plan, and we had no missing data. One limitation is that we did not use a cross-over, cross-back approach, which means that part of the findings may be explained by natural changes in respiratory physiology. For example, the changes in ΔP and MP over the hours we observed in the patients could also have been caused by an improvement in the clinical condition, independent of the way ventilation was applied. This, however, is very unlikely considering the fact that the overarching study findings showed only marginal changes in ventilator settings and parameters of interest over the first 4 days of invasive ventilation [12]. Last, we collected data only within a relatively short timeframe of 5 h, while in most COVID-19 patients with ARDS, liberation from ventilation lasts many days to weeks [12,17,18].

Our findings are in line with those from previous investigations testing the safety, feasibility and effectiveness of INTELLiVENT-ASV in different patient groups. In one randomized clinical trial in postcardiac surgery patients [19], INTELLiVENT-ASV resulted in less MP during postoperative ventilation. In another prospective observational study in a general ICU population [20], INTELLiVENT-ASV demonstrated a lower ΔP and less MP. We ourselves recently showed ASV, the predecessor of INTELLiVENT-ASV, to have comparable effects on the intensity of ventilation [21]. ASV uses the same algorithms as INTELLiVENT-ASV for adapting ventilator settings that affect both ΔP and MP.

The findings did not change in the sensitivity analysis using an alternate equation for calculating MP. The original equation is designed for use in volume-controlled ventilation, while we used pressure-controlled ventilation before changing the ventilator mode to INTELLiVENT-ASV, and INTELLiVENT-ASV itself is a mode that is based on the principles of pressure-controlled ventilation. Findings also did not change in a sensitivity analysis in which MP was normalized. The rationale behind normalization is that ventilator-induced lung injury derives from the interaction between the causal factors of two broad categories—the machine output and the condition of a healthy or diseased lung. This means that for the same amount of MP, the lung volume determines the intensity.

One key ventilator setting that showed a remarkable change from before to after the conversion to automated ventilation was the RR. Previous studies have shown an association of higher RR with poor outcomes [22]. The decrease in RR resulted in a lower minute volume, but etCO_2_ and SpO_2_ values were not affected. We hypothesize that the slight increase in V_T_ resulted in the recruitment of parts of the lung sufficient to compensate for the lower minute volume.

Compared to conventional ventilation, INTELLiVENT-ASV resulted in a somewhat higher median V_T_. However, in most patients, V_T_ remained largely within the widely agreed safety zone, i.e., <8 mL/kg PBW. The increased median C_RS_ with INTELLiVENT-ASV could mean that the larger V_T_ was applied to a better-aerated lung. The lower ΔP and lower upper airway pressure with INTELLiVENT-ASV are in line with this suggestion. With INTELLiVENT-ASV, median ΔP remained below the suggested safety limit of 15 cm H_2_O and was lower than with conventional ventilation in most patients. One interesting finding of our study was that when ΔP was high with conventional ventilation, V_T_ decreased with INTELLiVENT-ASV, suggesting less overinflation after the conversion. While we acknowledge the importance of using a low V_T_ in patients with ARDS, a small increase in V_T_ accompanied by decreases in ΔP and MP may be acceptable.

## 5. Conclusions

Conversion from conventional ventilation to INTELLiVENT-ASV resulted in a lower intensity of ventilation in this cohort of COVID-19 ARDS patients. The conversion resulted in a small but acceptable increase in V_T_, as V_T_ remained within the generally accepted safety limits. The effect of INTELLiVENT-ASV on MP seems mainly driven by a reduction in the respiratory rate.

## Figures and Tables

**Figure 1 jcm-10-05409-f001:**
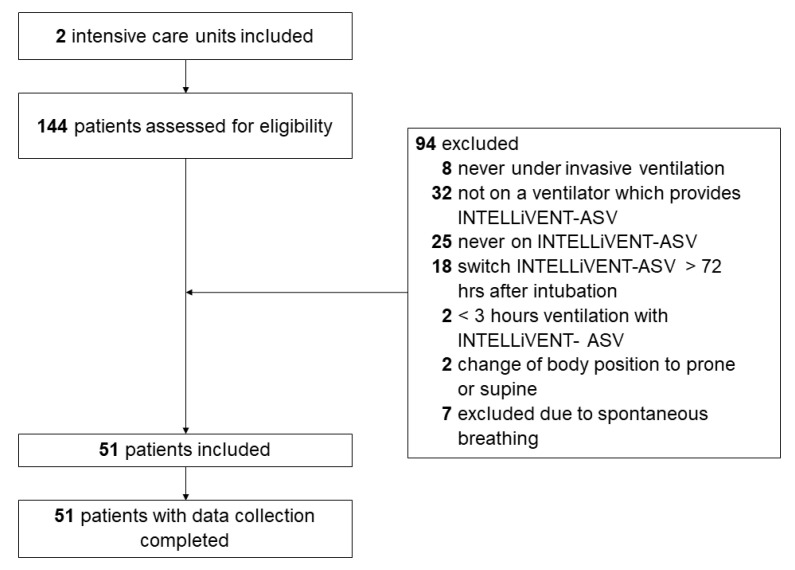
Study profile. Consort diagram showing flow of patients.

**Figure 2 jcm-10-05409-f002:**
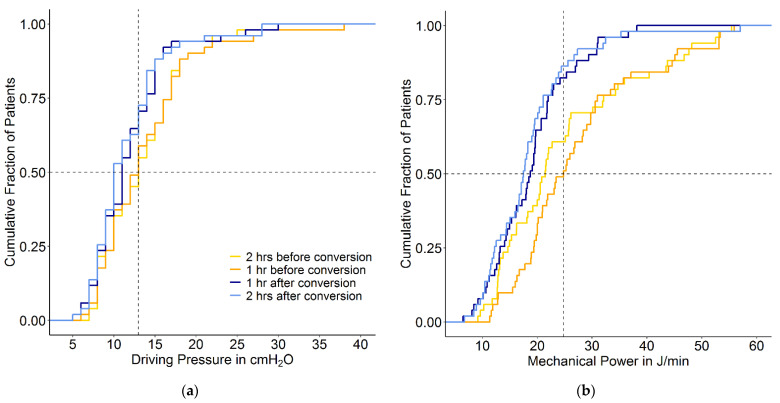
Cumulative frequency distribution of (**a**) ΔP and (**b**) MP. The plots show ΔP and MP 2 and 1 h before the conversion and 1 and 2 h after the conversion from conventional ventilation to INTELLiVENT-ASV. Vertical dotted lines represent the median at the last hour before the conversion, and horizontal dotted lines show the respective proportion of patients reaching each cutoff.

**Figure 3 jcm-10-05409-f003:**
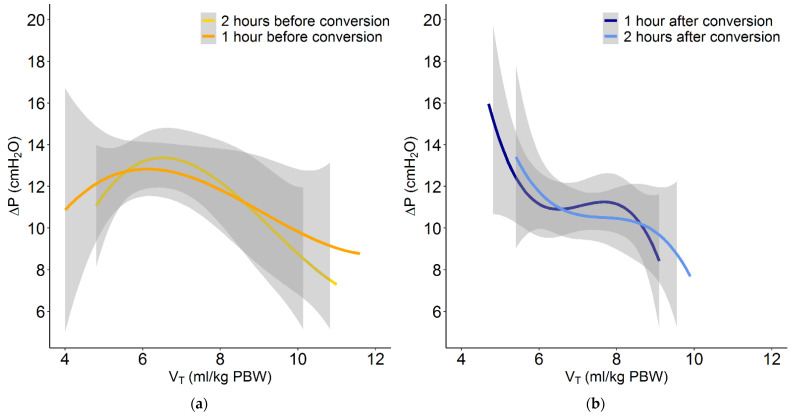
Plot of the relation between V_T_ and ΔP with conventional ventilation at (**a**) 2 and 1 h before conversion to INTELLiVENT-ASV, and at (**b**) 1 and 2 h after conversion.

**Figure 4 jcm-10-05409-f004:**
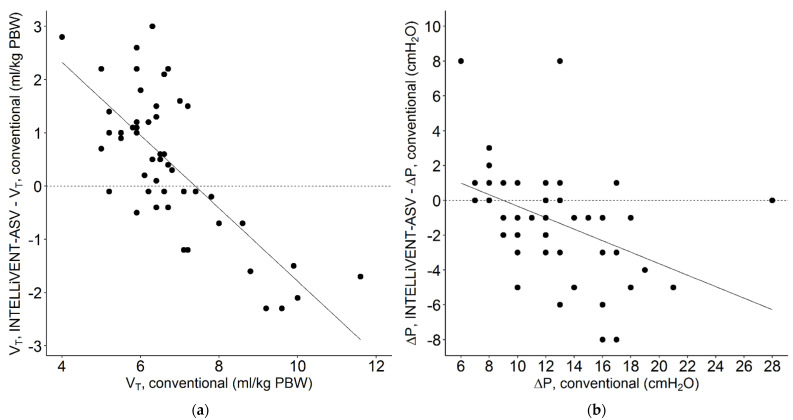
Scatterplot of individual changes in (**a**) tidal volume (ml/kg PBW) and (**b**) driving pressure (cm H_2_O) when the patients were switched from conventional ventilation, at 1 h before conversion, to INTELLiVENT-ASV (V_T_, INTELLiVENT-ASV-V_T_, INTELLiVENT-ASV; and ΔP, INTELLiVENT-ASV-ΔP, conventional) 2 h after conversion. Continuous line; regression lines. Each patient was characterized by a single data point. A negative value for V_T_ or ΔP means that the conversion to INTELLiVENT-ASV resulted in a lower V_T_ or ΔP, and a positive value means that V_T_ or ΔP increased after the conversion.

**Figure 5 jcm-10-05409-f005:**
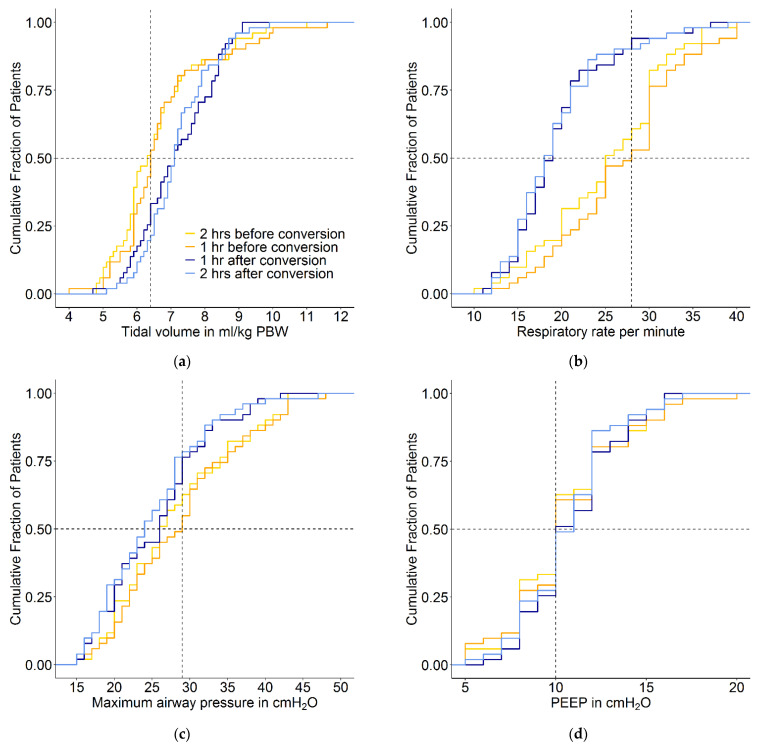
Cumulative frequency distribution of (**a**) tidal volume, (**b**) respiratory rate, (**c**) maximum airway pressure, (**d**) PEEP, (**e**) etCO_2_ and (**f**) SpO_2_. The plots show the ventilation variables 2 and 1 h before the conversion and 1 and 2 h after the conversion from conventional ventilation to INTELLiVENT-ASV. Vertical dotted lines represent the median at the last hour before the conversion, and horizontal dotted lines show the respective proportion of patients reaching each cutoff.

**Table 1 jcm-10-05409-t001:** Ventilator settings and limits.

	Conventional Ventilation	INTELLiVENT-ASV
Ventilator Settings		
etCO_2_ target	4.5 to 6.0 kPa	4.5 to 5.5 kPa
SpO_2_ target	90 to 92%	89 to 93%
Tidal volume target	5 to 8 mL/kg PBW	5 to 8 mL/kg PBW
Ventilator Limits		
Maximum airway pressure	30 cm H_2_O	30 cm H_2_O
PEEP limit	≤12 cmH_2_O	5 to 12 cmH_2_O
Tidal volume alarm limit	9 mL/kg PBW	9 mL/kg PBW
Respiratory rate limit	30 per min	30 per min
FiO2 upper limit	0.60	0.60

Abbreviations: etCO_2_, end-tidal carbon dioxide; SpO_2_, peripheral oxygen saturation; PEEP, positive end expiratory pressure; FiO_2_, fraction of inspired oxygen. With INTELLiVENT-ASV, the target ranges for etCO_2_ and SpO_2_ are pre-specified and set automatically; when a certain target is chosen, the lower and upper limits of the target ranges are 0.5 kPa under and above the target of 5 kPa for etCO_2_, and 2% around the 91% for SpO_2_.

**Table 2 jcm-10-05409-t002:** Baseline characteristics.

Characteristic	Specification	
Age, years		63 (51–69)
Gender	Men	39/51 (76)
	Women	12/51 (24)
Height, cm		178 (172–182)
Weight, kg		90 (77–103)
BMI, kg/m^2^		28 (25–32)
Position	Prone	19/51 (37)
	Supine	32/51 (63)
Administration of NMBA		17 (33)
Conversion to I-ASV, hours		10 (3–48)
Vital signs	Heartrate, bpm	93 (77–107)
	MAP, mmHg	75 (70–82)
	SpO_2_, %	92 (91–94)
Arterial blood gas	pH	7.39 (7.31–7.47)
	PaO_2_, kPa	9.2 (8.6–10.1)
	PaCO_2_, kPa	6.4 (5.5–7.1)
	Bicarbonate, mmol/l	26 (24–33)
	Arterial sat, %	93 (92–95)
PaO_2_/FiO_2_ ratio, mmHg		125 (95–165)
Severity of illness	APACHE IV	59 (45–70)
Severity of ARDS	Mild	7 (14)
	Moderate	28 (55)
	Severe	16 (31)
Chest CT-scan performed		28/51 (55)
Lung parenchyma affected	0%	3/28 (11)
	25%	7/28 (25)
	50%	7/28 (25)
	75%	9/28 (32)
	100%	2/28 (7)
Chest X-ray performed		41/51 (80)
Quadrants affected	1	6/41 (15)
	2	11/41 (27)
	3	14/41 (34)
	4	10/41 (24)
Co-existing disorders	Hypertension	18/51 (35)
	Heart failure	2/51 (4)
	Diabetes	9/51 (18)
	Chronic kidney disease	1/51 (2)
	COPD	3/51 (6)

Data are median (IQR) or N/total (%). BMI: Body Mass Index; NMBA: neuromuscular blocking agents; I-ASV: INTELLiVENT-ASV; APACHE: Acute Physiology and Chronic Health Evaluation; CT: computed tomography.

**Table 3 jcm-10-05409-t003:** Ventilation parameters at the predefined timepoints before and after the conversion from conventional ventilation to INTELLiVENT-ASV.

Parameter	2 h before Conversion	1 h before Conversion	1 h after Conversion	2 h after Conversion	*p* Value
∆P (cmH_2_O)	13 (10–17)	13 (10–17)	11 (9–14)	10 (8–14)	<0.001
MP (J/min)	21.5 (14.6–32.1)	24.8 (19.4–31)	18.8 (12.2–22)	17.5 (12.2–21.1	<0.001
V_T_ (mL)	450 (400–530)	473 (420–540)	516 (455–568)	520 (478–585)	0.008
V_T_ (mL/kg PBW)	6.3 (5.8–7.2)	6.4 (5.9–7.1)	7.1 (6.3–8.2)	7.1 (6.5–7.8)	0.008
PEEP (cmH_2_O)	10 (8–12)	10 (8–12)	10 (9–12)	11 (9–12)	0.5
Pmax (cmH_2_O)	26 (22–34)	29 (22–35)	26 (20–29)	24 (19–28)	<0.001
Pplat (cmH_2_O)	24 (20–28)	25 (21–28)	23 (18–25)	22 (19–25)	0.002
Pinsp (cmH_2_O)	14 (9–18)	14 (9–19)	12 (7–15)	11 (5–15)	<0.001
RR (bpm)	25 (20–30)	28 (22–30)	19 (16–21)	18 (15–21)	<0.001
Min. vol. (L/min)	11 (8.7–12.6)	11.8 (9.9–13.2)	9.9 (8.5–11.3)	9.8 (8.7–11.5)	0.002
FiO_2_ (%)	60 (40–70)	55 (45–65)	44 (35–56)	43 (32–53)	<0.001
SpO_2_ (%)	93 (92–96)	94 (92–96)	93 (92–95)	93 (92–94)	0.2
etCO_2_ (kPa)	6.1 (5.7–6.7)	5.9 (5.4–6.4)	5.9 (5.5–6.3)	5.9 (5.3–6.2)	0.5
C_RS_ (mL/cm H_2_O)	35 (25–49)	36 (26–53)	47 (35–63)	48 (35–63)	<0.001

Data are median (IQR). ∆P: driving pressure; MP: mechanical power; J/min: Joule per minute; V_T_: tidal volume; PBW: predicted body weight; cmH_2_O: centimeters of water; Pmax: maximum airway pressure; Pplat: plateau pressure; Pinsp: set inspiratory pressure; RR: respiratory rate; Min. vol.: minute volume; Bpm: beats per minute; FiO_2_: fraction of inspired oxygen; SpO_2_: pulse oximetry; etCO_2_: end-tidal carbon dioxide; kPa: kilopascal. C_RS_: compliance of the respiratory system.

## Data Availability

All data are available upon request.

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
