# Peer review of "Effect of INTELLiVENT-ASV versus Conventional Ventilation on Ventilation Intensity in Patients with COVID-19 ARDS—An Observational Study"

_jcm, 2021, doi:10.3390/jcm10225409_

Round 1
Reviewer 1 Report
In their manuscript “Effect of INTELLiVENT–ASV versus Conventional Ventilation on Ventilation Intensity in Patients with COVID–19 ARDS – an observational study”, Buiteman-Kruizinga and colleagues compared the intensity of ventilation (ΔP and MP) with INTELLiVENT–ASV versus conventional ventilation in a cohort of 51 COVID–19 ARDS patients in two intensive care units in the Netherlands. While certain limitations apply due to the small sample size as well as the retrospective nature of this study, the authors found that conversion from conventional ventilation to INTELLiVENT–ASV resulted in a lower intensity of ventilation. Furthermore, the authors conclude that these findings may favor use of INTELLiVENT–ASV in COVID–19 ARDS patients, but claim that future studies remain needed to see if the reduction in intensity of ventilation translates in clinical benefits in this set of patients.
While the study appears to be well planned and executed, the additional insights into what is already known about semi-automatic ventilation protocols are somewhat limited to the application of INTELLiVENT–ASV in Covid-19 patients.
However, this trial may inspire further research efforts in the field of automated ventilation protocols. A field that may only become more important with the ongoing labor shortage in critical care. Hence, this study will be of interest for the readership of JCM.
Author Response
Dear reviewer,
thank you for your comments, please see the attachment for the complete author's reply. Our comments at your review are written under 'Reviewer 1'.

Reviewer 2 Report
In this observational study, Buiteman-Kruizinga and colleagues reported a comparison between INTELLIVENT-ASV and conventional ventilation in COVID-19 patients. Specifically, they compared the intensity of ventilation by means of ΔP and MP in the two modalities.
INTELLIVENT-ASV has a great potential in adapting VT and RR to the mechanical properties of the patient respiratory system, as it continuously adjusts these parameters to minimize the energy applied to the system, according to the Otis equation. For example, if a low compliance is detected, the system lowers the tidal volume and increases the respiratory rate, thus reducing the ΔP (and likely the MP, due to the high “weight” of ΔP on MP calculation). An automated mode could be of use in a situation where hospital and staff’s resources are strained (e.g a pandemic) and frequent adjustments of ventilatory settings according to respiratory mechanics might prove challenging. However, we must stress the fact that this system does not behave properly in the presence of spontaneous activity, as compliance calculation becomes completely unreliable.
The study rationale is sound, the manuscript is well written, and the data presentation is original.
However, the study suffers from one major flaw that in my opinion precludes publication: the change of ΔP and MP is driven from a dramatic (and highly unlikely) change in compliance of the respiratory system (Crs, see Table 2). The latter changes from 26 (26-53) to 47 (35-63) from 1 hour before to 1h after the start of INTELLIVENT-ASV. The Crs almost doubles without either a change of PEEP (10>10) or recruitment maneuvers. This is unreasonable for any intensivist which deals with mechanical ventilation.
The study findings seem to suggest that (starting from a driving pressure of 13cmH2O) dramatic lung recruitment might be obtained by further increasing the tidal volume by 10-20%. This is very misleading and in contrast with the current strong evidence on ARDS patients, where lowering TV and ΔP is strongly associated with improved outcomes.
Author Response
Dear reviewer,
thank you for your comments, please see the attachment for the complete author's reply. Our comments at your review are written under 'Reviewer 2'.

Reviewer 3 Report
Buiteman-Kruizinga et al present results from a subgroup analysis of patients who received INTELLiVENT-ASV within an observational cohort study of mechanical ventilated COVID patients. The primary outcomes were mechanical power and driving pressure required during INTELLiVENT-ASV (hereafter referred to as ASV) compared with conventional ventilation. Both primary outcomes were improved in the ASV group, at the expense of tidal volume.
This is a well designed cohort study with a pre-specified analysis plan conducted at two centers with strong experience in providing lung-protective ventilation. Although the study period was relatively short, the time points of data collection were standardized. The finding that mechanical power and driving pressure were decreased is of interest because these variables are strongly suspected to impact risk of VILI, but titration using them as endpoints may be prohibitively work-intensive. I have several minor comments:
- Please provide P:F values for patients before and after conversion to ASV, if available, or baseline P:F in addition to stratifying ARDS severity by mild/moderate/severe.
- Please summarize ventilator modes used during conventional mechanical ventilation and the INTELLiVENT-ASV settings chosen (etCO2 target, O2 target, pressure limits).
- It is states on page 2, line 91 that patients were excluded from conversion to ASV if spontaneous breathing was detected at any time during data collection. How was this determined (analysis of ventilator waveforms? respiratory rate greater than set rate?) and achieved (deep sedation? only 33% of patients received NMB). Was spontaneous breathing allowed during ASV, and could this have affected measurements of plateau pressure?
- It is stated that there was an "acceptable" increase in tidal volume during ventilation with the ASV mode. It is unclear what acceptable means in this context, as data does not suggest a "safe" threshold below which tidal volumes are all acceptable-i.e. mortality increases with larger tidal volumes even within the usual target range of 4-8 cc/kg PBW (Needham D et al. BMJ Clinical Research April 2012 344(7854):e2124) and very low tidal volumes (3 cc/kg vs 6 cc/kg) may confer advantages in ventilator-free days (Bein T et al. , 10 Jan 2013, 39(5):847-856). The impact of larger tidal volumes over time is unclear from this study.
- Please summarize proportion of patients who received >8cc/kg PBW tidal volumes in ASV vs conventional modes.
Author Response
Dear reviewer,
thank you for your comments, please see the attachment for the complete author's reply. Our comments at your review are written under 'Reviewer 3'.

Round 2
Reviewer 2 Report
The Authors answered to all my critiques. I found the paper improved. I have no further comments.